# On the Evaluation of the Spin Galvanic Effect in Lattice Models with Rashba Spin-Orbit Coupling

**Götz Seibold** [1,*], **Sergio Caprara** [2], **Marco Grilli** [2] and **Roberto Raimondi** [3]

[1] Institute of Physics, Brandenburgische Technische Universität Cottbus-Senftenberg, P.O. Box 101344, 03013 Cottbus, Germany

[2] Dipartimento di Fisica Università di Roma Sapienza, Piazzale Aldo Moro 5, I-00185 Roma, Italy; sergio.caprara@roma1.infn.it (S.C.); marco.grilli@roma1.infn.it (M.G.)

[3] Dipartimento di Matematica e Fisica, Università Roma Tre, Via della Vasca Navale 84, 00146 Rome, Italy; roberto.raimondi@uniroma3.it

[*] Correspondence: seibold@b-tu.de

**Abstract:** The spin galvanic effect (SGE) describes the conversion of a non-equilibrium spin polarization into a charge current and has recently attracted renewed interest due to the large conversion efficiency observed in oxide interfaces. An important factor in the SGE theory is disorder which ensures the stationarity of the conversion. Through this paper, we propose a procedure for the evaluation of the SGE on disordered lattices which can also be readily implemented for multiband systems. We demonstrate the performance of the method for a single-band Rashba model and compare our results with those obtained within the self-consistent Born approximation for a continuum model.

**Keywords:** spintronics; spin-galvanic effect; lattice models

## 1. Introduction

Spin-orbit coupling (SOC) lies at the heart of spin-to-charge conversion [1], which is a major issue for future spintronics technologies [2,3]. Bychkov and Rashba [4] proposed that, in a two-dimensional electron gas (2DEG), the lack of inversion symmetry along the direction perpendicular to the gas plane leads to a momentum-dependent SOC usually described by the Rashba Hamiltonian

$$H = \frac{p^2}{2m} + \alpha \mathbf{z} \times \boldsymbol{\sigma} \cdot \mathbf{p},$$ (1)

where $\mathbf{p}$ is the momentum operator for motion along the 2DEG plane, say the $x$-$y$ plane, and $\mathbf{z}$ is a unit vector perpendicular to it. $\boldsymbol{\sigma} = (\sigma^x, \sigma^y, \sigma^z)$ is a vector of the standard Pauli matrices and $\alpha$ is a coupling constant whose strength depends on the SOC of the material and the field responsible for the parity breaking. In the last decade, progress in the technology of thin film growth has enabled the exploration of numerous phenomena where "Rashba Physics" plays a fundamental role (cf. e.g., [5,6]). With regard to spin-charge conversion, two major effects are discussed: the spin Hall effect (SHE) originally proposed by Dyakonov and Perel in 1971 [7], and the spin galvanic effect (SGE) which was first predicted by Ivchenko and Pikus [8] and later studied within the Rashba model by Edelstein [9] in its reciprocal manifestation (inverse SGE). The SHE and its reciprocal manifestation are the coupling between charge and spin currents flowing perpendicular to each other where the current spin polarization is perpendicular to both the charge and spin current's flow (see, e.g., [10,11]). Therefore, spin polarization due to an applied electric field accumulates only at the sample edges. In contrast, SGE and inverse SGE describe the interconversion of charge current and *bulk* spin polarization (see, e.g., [12] for a review) and has recently been observed to yield an

unexpectedly large spin-to-charge conversion at oxide interfaces [13–15], i.e., between $LaAlO_3$ and $SrTiO_3$. A minimal model for the electronic structure of such interfaces usually involves the Ti $t_{2g}$ orbitals [16]. The individual bands are further spin-split by atomic spin-orbit coupling and an interface asymmetry term [17,18]. Alternatively, the splitting can be analyzed using the $\mathbf{k} \cdot \mathbf{p}$ Luttinger-Kohn (LK) approach [19,20]. In both cases, this leads to a complex structure of the Rashba SOC (RSOC), which for the atomic SOC in combination with an interface asymmetry has been derived in [21,22]. The evaluation of the SGE response for such couplings on a lattice is a challenging task and requires a method which is also applicable in the presence of disorder. The standard impurity technique involves a three-step procedure: (a) calculation of the irreducible self-energy in the self-consistent Born approximation for the determination of the single-particle Green function; (b) calculation of the vertex corrections for the charge current by solving the corresponding Bethe-Salpeter equations; and (c) computation of the response function from a convolution of the Green functions and the vertices. This procedure has been successfully applied for the evaluation of the inverse spin galvanic effect [23], anisotropy magnetoresistance [24,25], and the spin Hall effect [26] in *single-band* Rashba models. However, similar investigations for multiband models with broken inversion symmetry have not been conducted yet to the best of our knowledge. In fact, while the general theory has been worked out (cf. e.g., [27] for the anomalous Hall effect), previous investigations have been restricted to inversion-symmetric multiband systems where vertex corrections vanish (cf. e.g., [28]). In fact, the increasing demand for the solution of the Bethe-Salpeter equations with increasing number of bands suggests the use of alternative approaches for the evaluation of the SGE response in disordered systems. One possibility would be the use of a multiband generalization of quasi-classical Green functions or its diffusive limit (in the context of superconductivity known as the Eilenberger and Usadel equations, cf. e.g., [29]). Alternatively, one could diagonalize directly the microscopic Hamiltonian on finite lattices and induce disorder by a suitable distribution of local and/or intersite potentials and evaluate the Kubo response function numerically from the eigenvalues and eigenstates. This approach has been previously followed in [22] for the evaluation of the SGE in a multiband model for oxide interfaces. Disorder was implemented via a flat distribution of local chemical potentials.

However, to substantiate the results of [22], it is necessary to demonstrate that, for a simpler single-band model, this method leads to results which agree with those obtained with the standard impurity technique mentioned above. This is precisely the purpose of this paper.

In Section 2, we introduce the necessary response function for the SGE which is subsequently analytically computed in Section 3 for the model Equation (1) both in the clean and disordered limit. In Section 4, we compare these results with the response functions obtained by diagonalizing numerically the corresponding lattice version of Equation (1) supplemented with local disorder. We conclude our discussions in Section 5.

## 2. Response Functions

The response function for the SGE is defined via [23]

$$\sigma_{SGE}(\omega) = (-e)\frac{\langle\langle J_x; S^y \rangle\rangle_\omega}{i(\omega + i\eta)} \tag{2}$$

with

$$\langle\langle J_x; S^y \rangle\rangle_\omega = \frac{1}{N}\sum_{k,p}(f_k - f_p)\frac{\langle p|S^y|k\rangle\langle k|J_x|p\rangle}{\omega + i\eta + E_p - E_k} \tag{3}$$

where $J_x$ is the total particle current and $S^y$ denotes the total y-polarized spin. In principle, all quantities have to be evaluated for vanishing particle-hole lifetime $\eta \to 0$. However, as will be discussed in Section 4, the numerical approach requires a careful evaluation of this limit.

Upon applying the identity

$$\frac{1}{(\omega + i\eta)(\omega + i\eta + E_p - E_k)} = \frac{1}{E_p - E_k}\left(\frac{1}{\omega + i\eta} - \frac{1}{\omega + i\eta + E_p - E_k}\right) \tag{4}$$

to Equations (2) and (3), the real part of the response function Equation (2) can be decomposed in a "Drude" and a regular part

$$\Re\sigma_{SGE}(\omega) = D_{SGE}\delta(\omega) + \sigma_{SGE}^{reg}(\omega) \tag{5}$$

with

$$D_{SGE} = \frac{-e\pi}{N}\sum_{k \neq p}\frac{f_k - f_p}{E_k - E_p}\Re\langle p|S^y|k\rangle\langle k|J_x|p\rangle \tag{6}$$

$$\sigma_{SGE}^{reg}(\omega) = \frac{-e}{N}\sum_{k \neq p}\frac{f_k - f_p}{E_k - E_p}\Im\frac{\langle p|S^y|k\rangle\langle k|J_x|p\rangle}{\omega + i\eta + E_p - E_k}. \tag{7}$$

Here, $p$ ($k$) refers to the quantum numbers which classify the single-particle eigenstates $|p(k)\rangle$ and eigenvalues $E_{p(k)}$ of the system. We have adopted the term "Drude" from its optical conductivity analogue since for the SGE it describes the non-equilibrium generation of an electrical current, i.e., an increase with time, in response to a uniform spin polarization $S^y$. Therefore, it is only finite for a clean system (see below), whereas it is expected to disappear in the presence of disorder.

We also need to compute the regular part of the optical conductivity

$$\text{Re}\,\sigma_{xx}(\omega) = (e^2)\frac{\text{Im}\,\Lambda_{xx}^{jj}(\omega)}{\omega} \tag{8}$$

with

$$\Lambda_{xx}^{jj}(\omega) = -\sum_{k,p}(f_k - f_p)\frac{\langle k|J_x|p\rangle\langle p|J_x|k\rangle}{\omega + i\eta + E_k - E_p} \tag{9}$$

and for definiteness we have specified both the applied electric field and the current to be oriented along the $x$-direction.

## 3. Analytic Evaluation of the SGE in the Clean and Dirty Limit for a Free Electron Gas with RSOC

Here, we consider the Hamiltonian for a free 2D electron gas in the xy-plane subject to RSOC so that Equation (1) reads

$$H = \frac{p^2}{2m} + \alpha(\sigma^x p_y - \sigma^y p_x) \tag{10}$$

which upon diagonalizing has eigenvalues

$$\epsilon_\pm = \frac{p^2}{2m} \pm \alpha p$$

and eigenvectors

$$|\pm\rangle = \frac{1}{\sqrt{2}}\begin{pmatrix} \pm ie^{-i\theta} \\ 1 \end{pmatrix} \tag{11}$$

with $\tan(\theta) = p_y/p_x$. The SOC splits the Fermi surfaces and corresponding Fermi momenta and density of states can be obtained from an expansion in $\alpha$ as

$$p_\pm = p_F\left(1 \mp \frac{\alpha}{v_F}\right)$$

$$N_\pm = N_0\left(1 \mp \frac{\alpha}{v_F}\right).$$

In the above, $N_0 = m/(2\pi)$, $p_F$ and $v_F = p_F/m$ are the density of states at the Fermi surface, the Fermi momentum, and the Fermi velocity, respectively, in the absence of SOC. Here, for the sake of simplicity, we are using natural units such that $\hbar = 1$.

*3.1. Clean Limit*

With the basis states, Equation (11), the particle and spin operators are represented as

$$\hat{J}_x = \frac{\partial H}{\partial p_x} = \frac{p_x}{m}\sigma^0 - \alpha\sigma^y \tag{12}$$

$$\hat{S}_y = \frac{1}{2}\sigma^y \tag{13}$$

so that evaluation of the corresponding matrix elements in Equation (6) yields

$$\langle s|S_y|s'\rangle = -\frac{1}{4}(s+s')\cos(\theta) - \frac{i}{4}(s-s')\sin(\theta) \tag{14}$$

$$\langle s'|J_x|s\rangle = \frac{p_x}{m}\delta_{s,s'} + \frac{\alpha}{2}(s+s')\cos(\theta) + \frac{i\alpha}{2}(s-s')\sin(\theta). \tag{15}$$

For the Drude response, only interband transitions $s \neq s'$ are relevant and, with $\Re\langle s|S_y|-s\rangle\langle -s|J_x|s\rangle = -\alpha\sin^2(\theta)/2$, one obtains from Equation (6)

$$\begin{aligned}
D_{SGE} &= \frac{e}{4\pi}\sum_s \int dp\, p\frac{f(\epsilon_s) - f(\epsilon_{-s})}{\epsilon_s - \epsilon_{-s}}\frac{\alpha}{2}\int_0^{2\pi} d\theta\sin^2(\theta) \\
&= \frac{e}{8}\sum_s s(p_s - p_{-s}) \\
&= -e\frac{\alpha}{2}\pi N_0 = -e\frac{\alpha}{4}m\,.
\end{aligned} \tag{16}$$

Similarly, the regular part—Equation (7)—can be evaluated as

$$\sigma_{SGE}^{reg}(\omega) = \frac{e}{32\alpha}\left\{\Theta(|\omega| - 2\alpha p_{F,+}) - \Theta(|\omega| - 2\alpha p_{F,-})\right\} \tag{17}$$

where $p_{F,\pm} = \sqrt{2m\mu + (m\alpha)^2} \mp m\alpha$ denote the Fermi momenta of the two Rashba split bands and $\mu$ is the chemical potential. It is then straightforward to show that Equations (16) and (17) obey the sum rule

$$\int_{-\infty}^{+\infty} d\omega\,\Re\sigma_{SGE}(\omega) = D_{SGE} + \int_{-\infty}^{+\infty} d\omega\,\sigma_{SGE}^{reg}(\omega) = 0 \tag{18}$$

which, similar to the optical conductivity, follows from the Kramers-Kronig relation between real and imaginary part of the response function Equation (3). However, in contrast to the optical conductivity, there is no diamagnetic term in the spin galvanic response so that the frequency integral over $\Re\sigma_{SGE}(\omega)$ vanishes.

*3.2. Disorder Limit*

In the presence of disorder, a stationary state can be reached so that the singular Drude contribution Equation (6) vanishes (see next section) and the zero frequency response instead is governed by a finite value of the regular part $\sigma_{SGE}^{reg}(\omega = 0)$. It can be calculated from standard impurity techniques which are based on a determination of the Green function within the self-consistent Born approximation and the implementation of current vertex corrections in the response function.

In the presence of white-noise standard disorder, the Green function $\hat{G} = G_0\sigma^0 + G_1\sigma^x + G_2\sigma^y$ reads

$$\hat{G} = \frac{G_+ + G_-}{2}\sigma^0 - (\sigma^x\hat{p}_y - \sigma^y\hat{p}_x)\frac{G_+ - G_-}{2}$$

where

$$G_\pm = (\epsilon - \epsilon_\pm(\mathbf{p}) - \Sigma)^{-1}, \Sigma^{R,A} = \mp \frac{i}{2\tau}$$

with $\tau^{-1} = 2\pi n_i N_0 u^2$, $n_i$ being the impurity density and $u^2$ the square of the scattering amplitude, describes the correlations of the disorder potential, $\langle V(\mathbf{r})V(\mathbf{r}') = u^2\delta(\mathbf{r} - \mathbf{r}')$.

The vertex equation for the particle current vertex reads

$$\hat{J}_x = \hat{v}_x + \frac{1}{2\pi N_0\tau} \sum_{\mathbf{p}} \hat{G}^R \hat{J}_x \hat{G}^A$$

where $\hat{v}_x$ is the bare current vertex and $G^{R(A)}$ denotes the retarded (advanced) Green's functions. Since vertex corrections do not modify the momentum dependence of the vertex [25,30], it is useful to write the full vertex as

$$\hat{J}_x = \hat{v}_{x,0} + \hat{\Gamma}_x$$

where $\hat{v}_{x,0}$ represents the momentum-dependent part of the bare vertex $\hat{v}_x$. The resulting equation for the vertex $\hat{\Gamma}_x$ reads

$$\hat{\Gamma}_x = \hat{\gamma}_x + \frac{1}{2\pi N_0\tau} \sum_{\mathbf{p}} \hat{G}^R \hat{\Gamma}_x \hat{G}^A \tag{19}$$

where the effective bare vertex $\hat{\gamma}_x$ is defined by

$$\hat{\gamma}_x = \hat{v}_x - \hat{v}_{x,0} + \frac{1}{2\pi N_0\tau} \sum_{\mathbf{p}} \hat{G}^R \hat{v}_{x,0} \hat{G}^A.$$

One finds that $\hat{v}_x$ is given by Equation (12) and $\hat{v}_{x,0} = \frac{p_x}{m}\sigma^0$. The effective bare vertex can then be shown to vanish

$$\begin{aligned}
\hat{\gamma}_x &= -\alpha\sigma^y + \frac{1}{2\pi N_0\tau} \sum_{\mathbf{p}} \hat{G}^R \frac{p_x}{m} \hat{G}^A \\
&= -\alpha\sigma^y + \frac{\sigma^y}{4mN_0}(p_+ N_+ - p_- N_-) \\
&= -\alpha\sigma^y + \frac{\sigma^y}{4mN_0}\left(4\frac{\alpha}{v_F}\right)N_0 p_F \\
&= 0,
\end{aligned} \tag{20}$$

so that Equation (19) is solved by $\hat{\Gamma}_x = 0$ and the SGE response can be evaluated with the standard velocity term $\hat{J}_x = p_x/m$,

$$\begin{aligned}
\sigma_{SGE} \equiv \sigma_{SGE}^{reg}(\omega = 0) &= \left(-\frac{e}{2\pi}\right)\sum_{\mathbf{p}} \mathrm{Tr}\left[\frac{\sigma^y}{2}G^R \frac{p_x}{m}G^A\right] \\
&= \left(-\frac{e}{2\pi}\right)\mathrm{Tr}\left[\frac{\sigma^y}{2}\sum_{\mathbf{p}}G^R \frac{p_x}{m}G^A\right] \\
&= \left(-\frac{e}{2\pi}\right)\mathrm{Tr}\left[\frac{\sigma^y}{2}(\alpha\sigma^y)2\pi N_0\tau\right] \\
&= (-e)(\alpha N_0\tau).
\end{aligned} \tag{21}$$

We thus obtain the following relation between the SGE response in the dirty and clean limit, Equation (16),

$$\frac{\sigma_{SGE}}{D_{SGE}} = \frac{2\tau}{\pi} \tag{22}$$

which in the subsequent section will be used to validate our numerical approach for the evaluation of $\sigma_{SGE}$ on a disordered lattice.

## 4. Evaluation of the Spin Galvanic Effect for a Disordered Lattice Model with RSOC

We consider a 2D lattice model (size $N_x \times N_y$) with RSOC

$$H = \sum_{ij\sigma} t_{ij} c_{i\sigma}^\dagger c_{j\sigma} + \sum_{i\sigma} V_i c_{i\sigma}^\dagger c_{i\sigma} + H^{RSO} \tag{23}$$

where the first term describes the kinetic energy of electrons on a square lattice (lattice constant $a$, only nearest-neighbor hopping: $t_{ij} \equiv -t$ for $|R_i - R_j| = a$) and the second term is a local disorder potential with a flat distribution $-V_0 \leq V_i \leq V_0$.

The last term is the RSO coupling

$$H^{RSO} = \frac{\alpha}{2} \sum_i \left[ \widetilde{j^y_{i,i+x}} - \widetilde{j^x_{i,i+y}} \right] \tag{24}$$

where

$$\widetilde{j^\nu_{i,i+\eta}} = -i \sum_{\sigma\sigma'} \left[ c^\dagger_{i\sigma} \sigma^\nu_{\sigma\sigma'} c_{i+\eta,\sigma'} - c^\dagger_{i+\eta,\sigma'} \sigma^\nu_{\sigma'\sigma} c_{i,\sigma} \right] \tag{25}$$

denotes the $\nu$-component of the spin-current flowing on the bond between $R_i$ and $R_{i+\eta}$. The tilde indicates that Equation (25) only represents that part of the spin current which is derived from the kinetic energy. In momentum space and in the basis of $(c_{k,\uparrow}, c_{k,\downarrow})$, Equation (24) can be written as

$$H^{RSO}_{kk} = \alpha \left[ \sin(k_y)\sigma^x - \sin(k_x)\sigma^y \right]$$

which for small momenta coincides with the RSOC in Equation (10).

Figure 1 reports the Drude SGE response obtained from Equation (6) for the clean lattice model as a function of chemical potential $\mu$, specified in units of the bandwidth parameter $B = 4t$. For small RSOC $\alpha/B$, the Drude response agrees with the expected result from the continuum model (symbols), Equation (16), with the mass given by $m = 1/(2t) = 2/B$. Deviations occur for large coupling and close to the band edges where only one of the Rashba split subbands is occupied. Major deviations occur also for large $\alpha/B$ and around $\mu = 0$, where $D_{SGE}$ changes sign due to the transformation of the dispersion from electron to hole-like. Note, however, that, due to particle-hole symmetry, the relation $D_{SGE}(\mu) = -D_{SGE}(-\mu)$ is always obeyed.

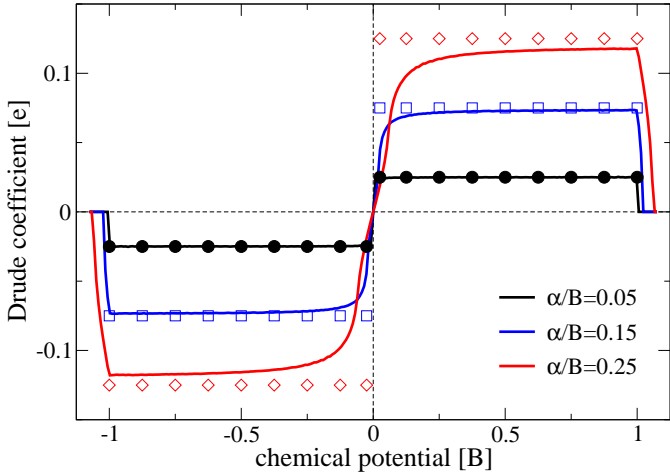

**Figure 1.** Drude coefficient of the spin galvanic response $D^{SGE}$ for the single-band Rashba lattice model as a function of chemical potential (solid lines). The symbols correspond to $D_{SGE} = -e\alpha/(2B)$ obtained from Equation (16) with the replacement $m = 2/B$ and the bandwidth parameter $B = 4t$.

We proceed by evaluating the SGE response for the disordered system and by validating our numerical results on the basis of the relation Equation (22). The latter depends on the momentum relaxation time $\tau$ which can be obtained by fitting the optical conductivity, obtained from Equation (8), with the Drude formula

$$\sigma(\omega) = \frac{\sigma_0}{1 + (\omega\tau)^2}. \tag{26}$$

The main panel of Figure 2 reports the numerical result (dots) for the optical conductivity which has been obtained by averaging Equation (8) over 50 disorder configurations and 50 boundary conditions along *x*- and *y*-directions on a $N_x \times N_y$ lattice with $N_x = N_y = 30$. The boundary conditions are specified by the phase $0 \le \varphi \le 2\pi$ which is acquired by the wave-function upon performing a translation with the linear dimension of the system $|\Psi(\mathbf{R}_i + N_{x,y} a \mathbf{e}_{x,y})\rangle = exp(i\varphi)|\Psi(\mathbf{R}_i)\rangle$. This procedure can be thought of as a sampling of momenta between the points $(n2\pi/N_x, m2\pi/N_y)$ with $n, m = 0, \ldots, N_{x,y} - 1$ which correspond to $\varphi = 0$. Therefore, an average over random values of $\varphi$ mimics the states of a larger system.

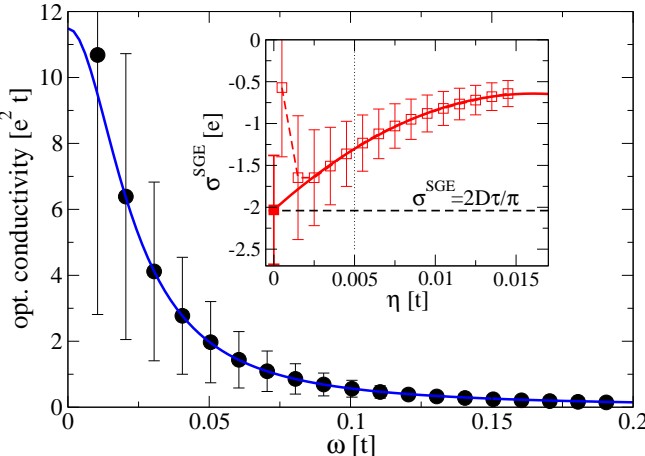

**Figure 2.** Optical conductivity (full dots) evaluated for disorder strength $V_0/t = 0.3$, chemical potential $\mu = -3t$, and RSOC $\alpha/t = 0.3$. The blue solid line is a fit to the Drude model Equation (26) with $\sigma_0 = 11.5e^2 t$ and $\tau = 43.81/t$. Inset: The SGE response Equation (7) as a function of the lifetime parameter $\eta$ for the same parameters. The red solid line is a polynomial fit to the data points with $\eta > 0.005$ corresponding to the average level spacing of eigenvalues (vertical dotted line). It extrapolates to a value of $\sigma^{SGE} \approx -2e$ in agreement with the value expected from the SCBA analysis (horizontal dashed line).

For the given parameters $V_0/t = 0.3$, $\alpha/t = 0.3$ and chemical potential $\mu = -3t$, the Drude fit yields a scattering time $\tau = 47.81/t$ and a DC conductivity of $\sigma_0 = 2.3e^2 t$.

Upon computing the SGE response from Equation (7), we are faced with the problem of always finding $\lim_{\eta \to 0} \sigma_{SGE} = 0$ for a finite lattice (cf. square data in the inset to Figure 2) where $\eta$ is the parameter which shifts the pole of $\sigma_{SGE}$ into the complex lower half-plane. We therefore adopt a method which has already been applied in [31] for the evaluation of the spin Hall coefficient and amounts to extrapolating $\sigma_{SGE}$ for values of $\eta$ larger than the average lattice level spacing $\delta$ until $\eta = 0$. For our $30 \times 30$ lattice, one has $\delta = 2B/(2 \times 900) = 0.0044t$ so that we perform the extrapolation for $\eta > 0.005t$ indicated by the vertical dotted line in Figure 2. As a fitting curve, we take a polynomial up to quadratic order. The resulting curve (solid line in the inset to Figure 2) extrapolates to a value $\sigma^{SGE} \approx -2e$ which is in agreement with the expected result Equation (22) from the continuum model (dashed horizontal line).

Figure 3a shows a comparison between the Kubo response calculation Equation (7) and the expected result from the continuum model Equation (22) as a function of chemical potential for $V_0/t = 0.3$ and $\alpha/t = 0.3$. Note that Equation (22) is evaluated with the Drude weight obtained for the lattice model (cf. Figure 1). One expects best agreement between both approaches at small band filling where both, continuum and clean lattice model, have circular spin-orbit split Fermi surfaces. Inspection of Figure 3a reveals that this is in principle correct although the data point obtained with the Kubo formula at the lowest chemical potential $\mu = -3.85t$ slightly deviates from the expected

behavior. We attribute this to the fact that, for some disorder configurations, one is already in the limit where only one of the subbands is occupied, leading to an effective decrease of $|\sigma_{SGE}|$, which explains the small deviation from the continuum result Equation (22).

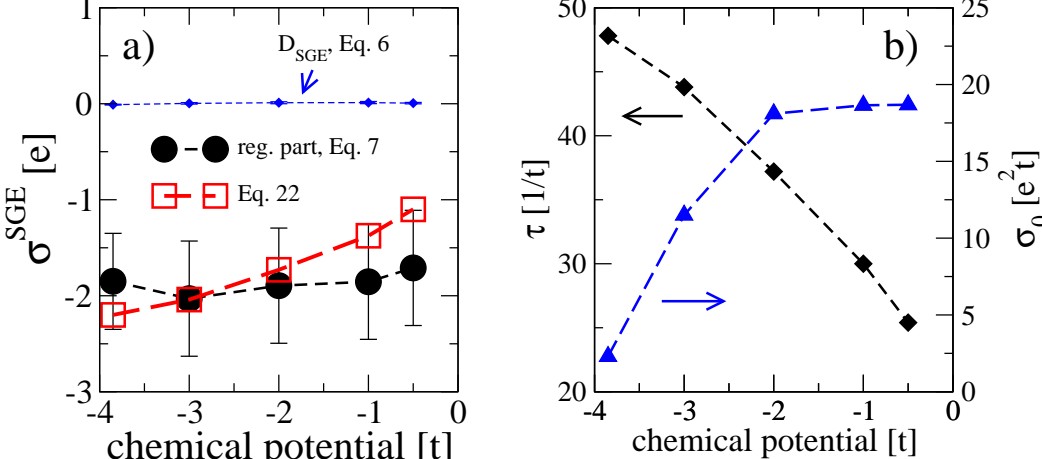

**Figure 3.** (**a**) Comparison of the SGE response evaluated with the Kubo formula Equation (7) and performing the $\eta$-extrapolation (squares) with the continuum result obtained from Equation (22). The SGE Drude coefficient $D_{SGE}$ (diamonds, error corresponds to symbol size) computed from Equation (6) is also shown. (**b**) The Drude formula parameters as a function of $\mu$ as extracted from fits to the optical conductivity. Calculations have been performed on $30 \times 30$ lattices for disorder potential $V_0/t = 0.3$ and RSOC $\alpha/t = 0.3$.

Finally, Figure 3a also reports the SGE Drude contribution from Equation (6) as a function of chemical potential. As anticipated, the $D_{SGE} = 0$ within the numerical accuracy due to the stationarity of the response in the presence of disorder.

## 5. Conclusions

We have discussed the problem of the numerical evaluation of the spin-galvanic response in a disordered lattice model. Such an approach has recently been used [22] for the computation of the SGE in a three-band model for oxide interfaces related to the observation of a large spin-to-charge conversion in these materials [13–15]. Two particular features could be reproduced in these investigations. First, at low temperature, a sign change of $\sigma_{SGE}$ is obtained when the chemical potential is at the Lifshitz point, i.e., where the $t_{2g}$ $xz$- and $yz$-orbitals become occupied. Second, at room temperature, the $xy$-orbitals yield a negligible contribution to $\sigma_{SGE}$ which only becomes significant when, upon gating, the chemical potential is tuned above the Lifshitz point [15]. The results obtained with the present approach fluctuate due to the finite size of the underlying lattice. An alternative would be the full solution of the Bethe-Salpeter equations for the disordered multiband interface system and the subsequent evaluation of $\sigma_{SGE}$ with the resulting vertex. Work in this direction is planned for the future.

**Author Contributions:** The authors equally contributed to this work.

**Funding:** G.S. acknowledges support from the Deutsche Forschungsgemeinschaft under SE806/19-1. M.G. and S.C. acknowledge financial support from the University of Rome Sapienza Research Project No. RM116154AA0AB1F5 and RM11715C642E8370.

**Conflicts of Interest:** The authors declare no conflict of interest.

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
