# Peer review of "On the Evaluation of the Spin Galvanic Effect in Lattice Models with Rashba Spin-Orbit Coupling"

_condensedmatter, doi:10.3390/condmat3030022_

Round 1

Reviewer 1 Report

In their manuscript, Seibold and coauthors investigate the spin galvanic effect (SGE) in a two-dimensional electron gas (2DEG) in the presence of Rashba spin-orbit coupling (SOC) and disorder. They compare an analytical derivation of the strength of the SGE with a direct numerical simulation, and find that they match quite well in the low-energy and low-SOC regimes. The SGE, which describes the conversion of charge to spin current, is of significant interest for the spintronics community, as spintronics devices require charge-to-spin and spin-to-charge conversion for the creation, manipulation, and readout of spin-polarized currents.

Despite this interest in the SGE, and in spin-to-charge effects in general, I am having difficulty seeing what is new or useful about this paper. The abstract and introduction say there is a need for a numerical method of calculating the SGE, but this has already been presented in their earlier work (their reference 22), and I find nothing new concerning the numerical method presented in this paper. In fact, the numerical section of the paper consists solely of the lattice discretization of the 2DEG+Rashba Hamiltonian, but this model was already developed many years ago. Meanwhile, although they do not explicitly state it, the numerical method appears to be just a direct diagonalization of this Hamiltonian. This is ok for small systems, but there are much more efficient approaches available for larger and more realistic systems. Finally, the results section appears to be just a validation of the numerical results, but this really is not surprising at all; of course the numerical and analytical results will match in the appropriate limits.

As it stands, I cannot recommend this paper for publication, as the numerical method is trivial and there is nothing new presented in the results. My recommendation is that the authors attempt to take this work beyond what is presented here, by using the numerical simulations to reveal some physics that cannot be so easily obtained analytically. This could be an investigation of different kinds of disorder, or perhaps a model other than 2DEG+Rashba.

Author Response

We thank the referee for his critical remarks which helped us to improve the presentation, in particular in the introduction, of our manuscript. We believe that most of the referees criticism is due to a poor motivation of our work which we have improved in the revised version (new text in lines 32-54). 

The referee is right that we have already applied the method in Ref. 22 for the evaluation of the spin

galvanic effect, however, its validity has not been shown which is the purpose of the

present paper. In fact, the agreement with the self-consistent Born approximation plus vertex corrections is by no means trivial since, as we discuss in our paper, the evaluation of the exact response function on a finite lattice with disorder in principle produces a "zero" result. Instead one has to consider the appropriate extrapolation of the response as a function of the particle-hole lifetime parameter. We demonstrate that this extrapolation converges to a result which agrees with the standard impurity technique when one extracts the momentum relaxation time from a fit of the optical conductivity to the Drude formula.

The referees statement that "the results section appears to be just a validation of the numerical results" is in this sense correct but the term "just" fails to recognize that this validation is non-trivial and a necessary prerequisite to apply the method to more complicated systems as done in Ref. 22

and which is of interest for the relevant community as is stated by referee 3.

Reviewer 2 Report

This manuscript presents a linear response analysis of spin galvanic effect in, possibly disordered, lattice tight binding models with  Rashba-type spin-orbit coupling (SOC). More specifically, the authors numerically evaluate the corresponding Kubo correlation function with а subsequent averaging over realizations of disorder. The results of a similar approach to a specific 3-band model of an interface band in some oxides have been recently reported in ref.22 by the same authors. Here the methodology and some technical subtleties are described in more details on a simple example of а 2d square lattice with Rashba SOC.

Apparently this paper is not a breakthrough, but it is a decent technical work which will be of interest for the relevant community. Therefore in my opinion this paper deserves publication after the authors will clarify/correct a few points in the presentation:

1. I believe it will be useful to add a brief explanation of how (4) is derived from (2). For example adding infinitesimal i\eta to \omega in the denominator in (2) will already make eq.(4) more plausible. What is the physical meaning of the ``Drude'' term in the context of the spin galvanic effect. I think it is important to explain this.

2. Normally, and in particular in Sec.3, p and k stand for momenta. In this context it would be in order to emphasize that in eq.(3-8) p an k are in general the quantum numbers classifying one particle eigenstates.

3. It is nor explained how the quantity defined in (6) is  related to eq.(4). Eq.(6) is just the steady state limit of the second term in (4), isn't it? Why not make the reading easier by stating this directly.

4. In Sec.3 in the clean limit the authors compute only the ``Drude weight`` D, but not the regular part \sigma_{reg}. Presumably the latter vanishes in the clean system. Why not demonstrating or at least stating this explicitly. The same concerns the dirty limit. How one can see that D vanishes, as the authors announce in Sec.2.

5. The spin current operator in eq.(2) is not fully defined as the authors do not explain what are the tau-tensors in the r.h.s.

6. When explaining their method that author state that in addition to the disorder realizations they average over ''50 boundary conditions``. What are specifically those boundary conditions. How they are defined and why it is necessary

Author Response

We thank the referee for his general appreciation of our work. In the following we give a detailedreply to the individual points.

1. I believe it will be useful to add a brief explanation of how (4) is derived from (2). For example

adding infinitesimal i\eta to \omega in the denominator in (2) will already make eq.(4) more plausible.

What is the physical meaning of the ``Drude'' term in the context of the spin galvanic effect.

I think it is important to explain this.

        --> We have revised the corresponding paragraph (lines 61-69) so that the

            derivation of the Drude and regular part is explained more clearly. This

            is also related to point (3) raised by the referee. We have also added

            a sentence to explain the meaning of the Drude term for the SGE.

2. Normally, and in particular in Sec.3, p and k stand for momenta. In this context it would be in order to emphasize that in eq.(3-8) p an k are in general the quantum numbers classifying one particle eigenstates.

        --> done after Eq. 6.

3. It is nor explained how the quantity defined in (6) is  related to eq.(4). Eq.(6) is just the steady state limit of the second term in (4), isn't it? Why not make the reading easier by stating this directly.

         --> see reply to point (1).

4. In Sec.3 in the clean limit the authors compute only the ``Drude weight`` D, but not the regular part \sigma_{reg}. Presumably the latter vanishes in the clean system. Why not demonstrating or at least stating this explicitly. The same concerns the dirty limit. How one can see that D vanishes, as the authors announce in Sec.2.

          --> The regular part does not vanish in the clean limit but only

              contributes at finite frequencies. In our revised manuscript we report

              the regular part in the new Eq. 17 and also discuss in the

              subsequent lines the sum rule for the spin galvanic response and that

              it is obeyed in the clean case. The vanishing of D in the dirty limit

              is more complicated. In fact, the approach we

              follow in Sec. III is only valid at low frequencies smaller than the

              interband excitations, however, the Drude part is exactly determined by

              the latter. We therefore demonstrate the vanishing of D within the

              numerical approach in the revised Fig. 3, Sec. IV and have added

              a corresponding statement in lines 152-154.

5. The spin current operator in eq.(2) is not fully defined as the authors do not explain what are the tau-tensors in the r.h.s.

          --> Probably the referee means (the new) Eq. 25. In fact this was a typos

              and we have changed 'tau' to 'sigma' which are the usual Pauli matrices

              already defined below Eq. 1. 

6. When explaining their method that author state that in addition to the disorder realizations they average over ''50 boundary conditions``. What are specifically those boundary conditions. How they are defined and why it is necessary

          --> We have added a corresponding explanation in lines 125-130.

Reviewer 3 Report

The paper considers the spin-galvanic response in a disordered lattice model based on the numerical evaluation within the Kubo response formalism. This paper is an extension of the previous paper of the authors, in PRL2017 (ref.[22]). The problem is related with the observation of a large spin-to-charge conversion in oxide interfaces (see ref.[13-15]), which attracts a large interest recently. The presented research reproduces a sign change of the spin galvanic response function when the chemical potential crosses the Lifshitz point in low temperature, the t2g xy- orbitals give significant contributions above the Lifshitz point.

The paper presents the new results in the modern and attractive field, and I recommend it to publish as it is.

Author Response

We thank the referee for her/his positive evaluation of our work and her/his recommendation for publication.

Round 2

Reviewer 1 Report

The authors have made a nice update to the introduction to better justify the numerical part of their paper. From their reply to my prior report, it is now more evident that a crucial aspect of the numerical methodology is to take the appropriate limit eta ---> 0. In my mind, this point should be emphasized more strongly. It is mentioned on line 63 and in the discussion around figure 2, but I think it should be reemphasized in conclusion, as it is an important detail for anyone who wants to use this approach on their own, and is necessary to obtain the correct result. With this minor update, I would be happy to recommend this paper for publication.